# Resources and Obstacles of a Maternity Staff Facing Intimate Partner Violence during Pregnancy—A Qualitative Study

**DOI:** 10.3390/healthcare11202782

**Published:** 2023-10-20

**Authors:** Yam Sureau, Marie-Rose Moro, Rahmeth Radjack

**Affiliations:** 1APHP, Hôpital Cochin, Maison de Solenn, 75014 Paris, France; marie-rose.moro@aphp.fr (M.-R.M.); rahmeth.radjack@aphp.fr (R.R.); 2Faculté de Médecine, Université de Paris Cité, 75000 Paris, France; 3GHU Paris Psychiatrie et Neurosciences, CPBB, 75014 Paris, France; 4Le Laboratoire Psychologie Clinique, Psychopathologie Psychanalyse (PCPP), Université de Paris, 92100 Boulogne-Billancourt, France; 5UVSQ, Inserm, CESP, Team DevPsy, Université Paris-Saclay, 94807 Villejuif, France

**Keywords:** intimate partner violence (IPV), gestational intimate partner violence (GIPV), healthcare professionals (HCPs), domestic abuse, pregnancy, trauma, assessment, empowerment, team support

## Abstract

Introduction: Intimate partner violence occurring during pregnancy has a similar prevalence as usual obstetrical disorders that are routinely screened for. Referenced publications insist on the importance of adequate screening, but the proper course of action has yet to be defined. Aim of study: We qualitatively explored the different resources and concepts that emerge from the discourse of maternity staff across professions. Material and methods: We led a semi structured interview with professionals, which included following their involvement with preselected patients. Nine professionals provided a sample of 19 interviews. The data was analysed using IPA methodology. Results We highlight the investigative importance of navigating the patient’s initial demand or lack thereof and the baby’s importance within, while identifying mechanisms of maternal disqualification. Creating an atmosphere prone to patient empowerment was the final theme to emerge from the study as the most beneficial tactic both in the short and long term. Conclusions: HCPs need to enable patients’ trust on a personal and an institutional level, as well as empowering the patient in the moment and respecting their values and choices. HCPs also convey the stability of the institution that has become a reference of refuge and assistance for patients from their pregnancy onwards.

## 1. Introduction

Intimate partner violence is an important public health issue with slowly progressing recognition. It overlaps several different medical specialties, with the psychiatric and psychotraumatic consequences on one side and the physical consequences on the other, mostly requiring the help of emergency-related healthcare divisions. Adverse effects of intimate partner violence occurring during pregnancy have been extensively reported on in the past literature [1,2], but there is little evidence regarding the specificities of tending to intimate partner violence sufferers in a maternity service. Fewer standard-of-care guidelines for the management of abuse cases apply to mother–infant specialties when compared to other divisions such as emergency services and psychiatry; as such, maternity health care providers (HCP) tend to rely on their own individual and collective experience in these abuse situations [3]. Reporting and analysing these personal resources and conducts, bearing such a crucial responsibility in both curative and preventive medicine, is the aim of the qualitative study we conducted in a Level 3 public maternity ward. French regulations categorize maternity hospitals into three tiers according to the clinical severity of patients they are able to treat, mostly according to the intensity of neonatal care they are able to provide based on gestational age. To qualify for the Level 3 tier, a maternity must have a neonatal ICU able to provide care for neonates under 33 weeks, and will often be the destination where at-risk pregnancies exceeding the resources of local level 1 and 2 maternities are transferred for care.

Intimate partner violence (IPV) is defined by the CDC (2020) as abuse or aggression (including coercive tactics) that occurs by a current or former intimate partner (i.e., spouse, boyfriend/girlfriend, dating partner, or ongoing sexual partner). This new terminology has the particularity of including former partners in its definition, which can be particularly relevant in a context of pregnancy, as opposed to “spousal” or “conjugal” violence that tend to solely focus on the current partner or require a specific marital status to be documented as such.

IPV can vary in how often it happens and how severe it is. It can range from one episode of violence that could have a lasting impact to chronic and severe episodes over multiple years. IPV can include several types of behavior: physical violence, sexual violence, stalking, psychological aggression (Table 1).

Gestational IPV (GIPV) are acts that occur during the perinatal period, which has different definitions according to the field of study; gynaecological or obstetrical studies tend to focus on the 9 months from conception to birth, whereas foetal medicine or paediatric studies usually focus on the second half of pregnancy (after 22 weeks). Nevertheless, psychiatric studies include the pre-conceptional period and first postnatal months as well, often rounding up the timeframe to the 12 months before birth to the first anniversary of the child’s birth [4]. 

Recent discourse has been implemented into the definition of GIPV events of reproductive submission, such as pregnancy coercion, interfering with birth control, and specific shame-based discourse, such as unfounded debate over the newborn’s paternity possibly threatening to withdraw child support and tamper with the mother’s social reputation [4].

The worldwide prevalence of GIPV in 2020 has been estimated at 3–9% in general population by several worldwide studies [2,5,6,7,8], reaching 9.2% in two 2021 worldwide review and meta-analysis [9]. These findings make GIPV as frequent in occurrence as the main obstetrical disorders routinely screened for, such as preeclampsia (1–5%) and gestational diabetes (2–8%) [5,6,10]. In high-risk populations that prevalence ranges from 13 to 71%, following the risk factors listed below [6,7,8,10,11]. This wide variation within populations is reported evenly throughout worldwide populations, and a large number of nation-based articles (India, Egypt, Ethiopia, Western Africa, Ireland, Vietnam, etc.), focus on their specific pregnant population [12,13,14,15]. Studying specific groups that differ in age, ethnicity, rural or urban living, likelihood of employment, marital status, access to contraception, confirm these findings when comparing these population groups. 

## 2. Current Evidence

The most common risk factors for GIPV are being unmarried, young age (16 to 29), low income, celibacy, lack of social security coverage, precarious living conditions, and those factors are consistent through worldwide high-impact publications [2,6,9,11,16]. 

A link between GIPV and unplanned pregnancy has been established in previous literature [2,11], with 40% of women asking for pregnancy termination reporting IPV during the previous year, and 19% of women undergoing GIPV describing pregnancy coercion or contraceptive sabotage on behalf of their partner [2,4,5,10]. Psychiatric illnesses have also been reported as correlating with occurrence of GIPV, for example, depression multiplying the risk of GIPV by a factor of 3 and substance abuse by a factor ranging from 2 to 6 depending on social subgroups in different studies [5,6,8,9,12]. Correlation between pre-existing PTSD arising from past trauma and the onset of GIPV has also been established, with some hypotheses as to its nature, namely that traumatic experiences have emotional and behavioural consequences that alters the victim’s relationship to others or coping mechanisms that can be weaponized by perpetrators to further their abuse, such as self-isolation, self-deprecation, fear, and isolation tendencies resulting from mistrust [1,2,12].

In 85% of GIPV cases, it occurs in a context of preexisting violence from the same perpetrator [4,6,7,12,16]. Cases of first occurrence can sometimes be described as a switch from an insidious act of violence to a more obvious one, with sexual violence as a prime example where a partner can escalate their pattern of violence from normalized intercourse coercion to physical sexual abuse. In other cases, new-occurring IPV is often linked to the couple’s level of agreement when it comes to the pregnancy itself. 

During pregnancy in the general population, violence tends to decrease during pregnancy, from 4.7 to 3.7% [2,8,17]. In the previously cited at-risk subgroups (younger in age, or bearing one or more social vulnerability factors), this tendency is actually inverted [5,6,9,11,17]; 71% of patients declared an increase in the severity of abuse, which often correlates with a potential disagreement within the couple over the desired nature of the pregnancy. Previous studies pondered different explanations regarding the divergent tendencies in IPV evolution during pregnancy, citing risk factors for worsening the abuse, such as potential sexual frustration, increased paranoid-like distrust over a sudden social surge in attention towards their victim, thus threatening the codependency or isolation pattern, or even gender dissatisfaction concerning the future child. 

Previous publications [1,2,17] established that GIPV has obstetrical, maternal, neonatal and paediatric consequences with a reliable homogeneity [1,5,6,7,9,10,16,17]. Chronologically, starting with the early prenatal period, GIPV is associated with an increased risk of miscarriage, frequency of UTIs (potentially related to unwanted intercourse frequency) [1,2,12]. Psychological GIPV has a specific association with delayed access to pregnancy care and insufficient weight gain in pregnant women. From a foetal standpoint, the risk of in utero foetal death is multiplied by 3, the risk of intrauterine growth restriction multiplied by 4, the risk of prematurity multiplied by 5, with statistical significance of overall violence, but also physical violence as well as isolated psychological abuse [1,2,5,6,12,17]. It is important to note these complications tend to be even more severe in the case of simultaneous substance use such as tobacco, cannabis or alcohol, which as stated previously, is in itself linked to the occurrence of IPV. During childbirth, gestational abuse has been significantly associated with adverse events such as placental abruption (RR = 5), perpartum hemorrhage (RR = 4), premature membrane rupture (RR = 8) [1,2,5,12]. Psychological abuse has also been significantly linked to acute respiratory distress in neonates [17]. 

Consequences of GIPV extend way beyond the perinatal period, and even though the potential ulterior exposure to violence in these children, be them witnesses or victims is hard to extricate, the level of increased risk for adverse events still seems relevant to mention [18]. At age 10, the risk of psychiatric diagnosis of any sort in these children is multiplied by 2.5, with a predominance in emotional and behavioral dysregulation [19] as well as speech disorders and sleep disorders. High blood pressure has also been reported in these children, as well as a threefold increase in the likelihood of developing asthma. 

In mothers, there is an expected increased risk of developing clinical depression [8], of which half includes PTSD symptoms [2,5,6,8,12,17]. The risk of substance abuse doubles with no decrease reported over time [12]. In the most extreme cases, it is been reported that 54% of maternal suicides follow events of GIPV. In the current social context of liberating speech regarding spousal homicide, 43% of female murder victims die at the hands of their partner during the perinatal period after suffering abuse during pregnancy [2,5].

## 3. Importance of a Prevention-Based Global Approach

The consequences of GIPV are well described in scientific literature, namely the consequences on maternal, fetal and long-term pediatric health. Nevertheless, this information is not routinely accessible to all patients that enter a maternity ward, and they may not always be accessible to such person, as well as seemingly unrelated questions as opposed to when seeking care for obvious abuse-related injuries. Added to the well described dire outcomes of GIPV in the scientific literature, which makes the caregiver’s practice all the more challenging, it is important to bear in mind the urgency of adapting monitoring and care of potential adverse events, while making sure the patient stays safe and comfortable within her care system [10,16,20,21]. 

A small number of qualitative studies have underlined the mainframe of maternity staff experiences in these situations, citing the individual sense of responsibility as a key element of care quality, as well as the importance of collective resources in a setting where theoretical education and specific training is described as insufficient. Additionally, the importance of establishing precise strategies has also been mentioned, in relation to the incentive of preserving the safety of the patient in a vulnerable position such as pregnancy [20,22].

During pregnancy, when a woman’s access to healthcare is routinely more frequent and encouraged by their surroundings, screening for IPV is all the more crucial [10,16,20], for different reasons; first, because the regularity of care might allow for a stronger trust between the patient and her healthcare provider (HCP) during the length of the pregnancy; second, because of the potential health risks for their unborn child, which can create a sense of urgency in protecting their child-to-be when they would previously not have protected themselves. It is common knowledge that, to this day, IPV is still vastly underreported, due to the potential social stigma, cultural or traditional belief systems, self-preservation instincts, lack of support systems or any other psychosocial vulnerability. This makes the role of health care providers all the more crucial in handling the care of a potential abuse victim, since they often end up being the first receiver of such life-altering revelations [3,10,16,21].

## 4. Aim and Value of Study

Adequate screening for GIPV thus requires specific areas of vigilance in every healthcare provider, having to watch out for signs of psychological unwellness [5,10,21], potential obstetrical pathologies, and the socio-familial red flags that would require the involvement of protective services. This intensity of awareness is only partly covered by the screening tools currently in place in public health services, namely direct questioning, use of visual tools such as the French «Violentometer», and especially bullet point questionnaires, which in waiting rooms put the patient at risk for spousal discovery and later retaliation, or simply might not be accessible to patients because of the language or literacy barrier. Overcoming those unchangeable issues relies solely on the practicioner’s ability to assess a situation and find an appropriate strategy for addressing this topic with a comprehensive and respectful attitude, in a trust-inducing setting, curated to the patients’ personal needs, opinions, beliefs and understanding. 

The limitations of such screening are multiple and sizeable. Finding an adequate timing and setting for entering this dimension of questioning can already pose a challenge, being sometimes faced with difficulty to see the patient alone without their spouse or interpreter in case of a language barrier. Furthermore, the eventual need for reiteration calls for a skilful and subtle demeanour in bringing a previously negated topic back up, risking potential patient distrust or feeling of intrusion, while constantly adapting their course of speech to respect the patient’s personal stance on the issue, and informing of potential bad outcomes of prolonged abuse during pregnancy. 

Few studies focus on the actual human involvement of HCP faced with such issues notably individual sensitivity, recognition of red flags such as loss of contact or insufficient checkups, the personal challenges of overcoming social stigma and respecting patient’s choice, the awareness of what existing resources can be deployed to each patient, symptom-based mental support, adequate teamwork with social and mental health services, adaptability to severity and level of urgency, and lastly the ability to extend a feeling of trustworthiness of the healthcare system beyond pregnancy, for whenever the patient feels ready and willing to seek change in an abuse situation [3,20,21,22]. Finnbogadóttir et al. (2020) conducted a content analysis, qualitative study that further outlined the personal feeling of responsibility from midwives’ experience when facing GIPV issues, and the very individual nature of both resources and barriers that arise.

Our focus is therefore to explore the humane, comprehensive, individual involvement elements of efficient harm reduction in situation of GIPV. The French healthcare system favors inpatient childbirth over home births, and prenatal follow up is put in place as soon as the patient registers for future birth in said maternity ward. Included in that follow up are regular ultrasound and bloodwork checkups performed by medical residents or midwives, and these appointments are often an occasion for the healthcare providers to screen for any potential elements that might call for the social worker or psychologist’s intervention. Simultaneously, prenatal care allows for anticipated registration with the local PMI (Mother-Infant Prevention center), which is a free, socially inclusive and neighbourhood-based service including midwives, nurses, mental health HCPs, and often part-time doctors, offering neonatal care in the local center or during house visits.

The aim of our study is to shed some light on the experience-based and individual resources different healthcare workers put to use in regard to screening for and taking on cases of GIPV. A secondary aim is to enable interpersonal dialogue and experience sharing around helpful strategies, as well as identifying the potential limitations, difficulties and qualitative experience triggered by those cases from a healthcare worker’s perspective. 

## 5. Materials and Methods

Our study consists of a qualitative analysis of caregivers’ perceived resources and limitations in their recollection of handling cases of GIPV using a semi-structured pre-written interview. The study was conducted in a public health service maternity ward. We identified the main professions of interest to be obstetrics and gynaecology physicians (OBGYNs), psychiatrists, psychologists, paediatricians, midwives, social workers and paediatric nurses. Rounding up to five HCPs on average per patient, we assessed that focusing on fve cases with an average of two participants per case would provide a sufficient sample of 10 interviews overall. Our monocentric design implies that participants might cross over between the cases; in order to provide the most case-specific data, we divided the questionnaire into two separate sections, the first being more focused on the interviewee’s overall experience and the other specifically relating to the case at hand (Table 2).

In order to obtain the most diverse array of personal experiences, we preemptively selected separate types of GIPV situations within the pool of patients whose cases were discussed in a bimonthly interdisciplinary medical and psychosocial meeting in a Level 3 maternity ward in Paris, France, from November 2022 to March 2023. Our study was conducted in a Level 3 centre, which allowed us to select cases of any clinical severity. We intended on including one case of confirmed violence with intent to separate from the perpetrator, one case without such intent, one case of suspected but unconfirmed GIPV, one case with prior familial violence unrelated to the current pregnancy, and one case of teenage pregnancy given the higher prevalence in younger women from age 16. 

Patients were selected after their staff presentation, solely on their status in regard to spousal violence independently from the potential complications arising from pregnancy in their cases. Access to patient files was granted once to the investigator for baseline data collection and inventory of healthcare workers’ identity for each case. 

Having selected those cases, we used patient files to obtain the names of the different workers present at the meeting, namely obstetricians, psychiatrists, paediatricians, midwives, psychologists, nurses, social workers, and reached out to them individually, in relation to each specific case with a separate email, asking for their voluntary, anonymized participation in a guided interview about that case. The interview questions were pre-written following a qualitative design, with open ended questions regarding the interviewee’s overall experience and sensibilities on the matter of GIPV, as well as their recollection of the specific patient situation and care, with Likert scales to visually assess their level of satisfaction regarding outcomes and quality of care among others (Table 3, Figure 1). The questionnaire was not made available to the healthcare workers beforehand, and the recorded interview results were triangulated with another study investigator so as to retrieve the most accurate and pertinent qualitative data.

In order to remain aware of each interviewee’s state of mind at every turn of the chosen situations, the participants were asked to answer questions on a Likert-inspired analogic visual satisfaction scale, providing additional insight into the participants’ step-by-step recollection of the situation. 

The questions asked on the visual scales were: “In my opinion…”, to which the participants provided an answer by placing a marker on a colored arrow as displayed below (Figure 1).

## 6. Ethical Design

Patient files were accessed on site by the investigator on one occasion. The MPS staff coordinator was present during the entire session and supervised the immediate anonymization of data as well as the relevance of every piece of information relevant to the study. Previous authorization had been obtained in writing from the maternity ward chief of surgery, and verbally confirmed a second time at the beginning of the data collection session. 

Participants were reached out to individually through their professional email addresses, providing them with a description of the aim of the study as well as the name of the patient they were solicited about, and built our participant sample on a voluntary basis. Written consent from willing participants was systematically collected and the anonymous nature of data analysis was specified before every interview. Interviews were recorded for further analysis, and recording files bore no mention of the participant’s identity nor the patient’s. 

## 7. Selected Cases Information

Patient data is summarized in Table 4, Table 5 and Table 6. As stated above, we included patient 1 as an abuse victim who had fled her husband’s recent abusive acts upon discovery of her pregnancy. Patient 2 was included on the grounds of severe past history of violence with no insight on current events. Patient 3 was included because of the intense warning signs of IPV as well as explicit coercion and submission. Patient 4 was included as a case of open ambivalence regarding the future of her relationship with the pregnancy’s father and perpetrator of IPV. Finally, patient 5 was included as an example of intrafamilial violence occurring during a teenage pregnancy.

## 8. Participants

The timeframe of our study allowed for the recruitment of nine healthcare providers, of whom two were social workers, two were psychologists, two were midwives, one was a pediatrician, one was an OBGYN, and one was a nurse. The inclusion criteria were mentioned in the patient records as having conducted one or more consultations with the selected patients, and reachability through the intra-hospital encrypted emailing system. 

In cases where two patients were treated by the same participant, two different interviews were conducted, but the generic questions part was not repeated a second time. This allowed the overall number of interviews to attain 8 generic interviews and 11 patient-based interviews, to give us 19 overall, which met the foreseen expectations required for qualitative significance in our IPA design. One HCP’s generic interview was not included in the study due to audio recording failure. The participant characteristics and frequency of interaction with each patient is mentioned in the table below (Table 7).

## 9. Analysis Methods

### 9.1. Analysis of Likert Scale

Given the amount of information gathered through the rest of the questionnaire, it seemed interesting to summarize the visual scale results into bar charts (Figure 2, Figure 3, Figure 4, Figure 5 and Figure 6) in order to notice, in one glance, the points of convergence and discrepancy between healthcare providers on each situation according to the question asked. Furthermore, the charts also highlight the specific areas where workers’ opinions and perception diverged, allowing for a more precise and pertinent understanding of the saturated results emanating from the integrated analysis.

### 9.2. IPA

The recorded interviews were analyzed using interpretative phenomenological analysis (IPA). IPA IPA takes into account the researcher’s subjectivity throughout the study, which cannot be overlooked in a case where the interview questions, conduction and result extrapolation were led by the same person. 

IPA is a particularly relevant qualitative approach for individual and integrative understanding of not only the interviewee’s experience, but also their personal takeaway after the fact and the diverse mental pathways put in place to make sense of every stage of the situation described, identifying the factors at play in both their assessment, demeanor and ulterior introspection regarding quality of the care they dispensed.

In order to maximize the specificity of patient-related answers, we started by analyzing each generic question separately in order for the phenomenological themes to emerge solely from patient-based experiences; next, the generic answers were integrated into those themes during rereadings so as to finally to attain data saturation and assess the reliability of convergence points.

## 10. Results

### 10.1. Likert Scale Answers

In order to better understand each caregiver’s state of mind when reminiscing upon the chosen situations, we used the previously described visual scales to precisely envision their level of satisfaction, overall experience and ulterior preoccupation. The results have been translated into 1–10 numerical values, as outlined in Table 8.

Given the amount of information gathered through the rest of the questionnaire, it seemed interesting to summarize the visual scale results into bar charts (Figure 2, Figure 3, Figure 4, Figure 5 and Figure 6) in order to notice, in one glance, the points of convergence and discrepancy between healthcare providers on each situation according to the question asked. Furthermore, the charts also highlight the specific areas where workers’ opinions and perception diverged, allowing for a more precise and pertinent understanding of the saturated results emanating from the integrated analysis. 

### 10.2. IPA Results

The first aim of our study was to isolate the participants’ personal strategies and warning signs when it came to addressing the question of GIPV for the first time, which are described in the first section. The three metathemes identified through the IPA analysis are reported on in the following sections.

#### 10.2.1. Addressing the Issue

Many participants were able to provide a go-to phrasing for their initial questioning regarding the potential presence of violence. Some of them expressed their habit of formulating their enquiry in a very direct manner, a tendency that proved to be nonspecific in regard to the profession.


*A: I don’t beat around the bush, I ask clear questions. Have you ever suffered physical or psychological acts of violence? Oftentimes, when I say psychological they ask me to explain what I mean by that, which is when I bring up the children if they have any, like have they ever seen anything… in any case, I make it really clear.*



*I: I always have if they’ve ever experienced any type of violence at any point in their life. I think I ask the question calmly, looking the patient in the eyes, I take my time.*


Others reported choosing a looser approach by initially investigating the patient’s relationship, entourage and feeling of security in the wake of the upcoming childbirth. 


*B: Usually I start with a much broader question, I ask the patient if she has a support system to rely on during the pregnancy and after birth*



*D: It is like dropping a stone into a deep well and looking for the echo, simple things such as “how are things with your partner”, getting them to open up about the relationship history which makes sense when discussing pregnancy.*


In both cases, over half of the participants reported engaging in a “double strategy”, where they would initially ask a direct question either focused on experienced hardships, or a broader enquiry into their living situations, then later in the consultation, circling back to the potential notion of violence through pregnancy-based questions, asking about their spouse’s demeanor, support, and what the patient imagines or expects from their living situation after leaving the maternity with the newborn. 


*E: So I enquire about both past and present abuse. But I have a double-edged strategy, that starts with these most generic questions and then at some point goes into how secure the patient feels when thinking about going back home with the baby, is the father going to be supportive, if there are highs and lows, how far do the lows go…*


The interview also took into account the possibility of patient reluctance or hostility towards such a question; some HCPs reported a generalized trust in the patients’ answers, or perceivable telltale signs of inauthenticity: 


*I: Sometimes the answer lies in how people answer the question. Patients who answer yes to being victims of abuse usually take time to reflect on the question, whereas patients who say no can sometimes be a bit too quick, too abrupt, it is like a “no” of convenience.*


But patients questioning the relevance of this topic within their journey of care seemed to be more frequent. In those cases, most participants shared phrases they tended to use in order to explain or justify the pertinence of asking about IPV, or opened up to the potential benefits of sharing this information with a professional. In most cases, patient reassurance seemed key to the success of those consultations, and by clearly expressing this to the patient, the dynamic of respect and service of the healthcare provider was often shared as being beneficial to the patients.


*B: What I say is, “I’m asking all those questions because it is important for me to get to know you so that I can better support you in your administrative endeavours, you don’t have to answer if you don’t want to and in no way am I trying to nag or pry, nor to make you uneasy in any way, really my purpose is to be as efficient as possible while assisting you*



*E: “Through my eyes, it seems like you’re going through something that can have many negative consequences on your health, and I believe the best thing to do to maintain your health would be (…), but I can’t put myself in your shoes, these choices are yours to make and whenever you feel good to go, if ever, we would help you through it, but it would be your own decision.*


#### 10.2.2. Warning Signs

Regarding the presence of warning signs or elements of past history that stressed the importance of screening for IPV, most of the participants shared their own internal “red flags”, but those appeared to be more profession-specific, i.e., linked to each person’s professional role towards the patients and what topics or symptoms are more likely to be addressed in each specific consultation.


*F: The pain in the a** patient, the one everyone calls insufferable, who’s always asking for something, in the emergency room every five minutes, with demands that don’t make sense… or the one who’s always late.*



*E: Patient with a history of preterm labor, frequent bleeding, small babies at birth, or pregnancies that weren’t followed up on well enough or the classic late term pregnancy discovery. Patients with plenty of chronic illnesses that aren’t properly monito red, or patients labeled as “psychiatric” even though nobody was able to figure out an actual diagnosis.*


It appeared that, in many cases, the first warning sign was obtained not by initial patient observation, but by information sharing within a team. This allowed many participants to underline the importance of adequate information sharing within each patient’s team in order to facilitate care and spare the patient from repeated questioning on the same, potentially triggering topics. 


*B: It is very precious that the resident sounded the alarm as early as she did, I think she was accurate picking up on signs of vulnerability*



*E: Sometimes workers are obligated to start over and tackle the question with a patient when they’ve already been briefed on the situation by previous colleagues, which is not that easy*


#### 10.2.3. Finding Support and Betterment through Teamwork

Teamwork was also reported as being essential in cases where notions of IPV had to be handled, namely because division of labor, according to each caregiver’s field of expertise, provided each team member with a clearer frame for their own mission regarding the patient, allowing for more personal confidence in the quality of their work as well as trust that every aspect of the question would be tackled by the best possible interlocutor in terms of helpfulness. 


*C: I didn’t go into too much detail because the social worker had just been there and I know she had formally investigated the spousal violence aspect, all of that.*



*I: I had asked around beforehand, I had gone to see the psychologist and social worker to learn about what to and not to say, so her care protocol could be as homogenous as possible and also so that we could somehow officialise what she was saying to us*


The crucial importance of teamwork was not limited to the necessity of adequate and timely transmissions, but was almost ubiquitously cited as a primordial resource for field-based peer learning through shared experiences. 


*C: We had lawyers come to the maternity ward, explaining proper protocols, giving us insight on how certain situations should be handled, so we would be more at ease*



*D: I was lucky enough to work with highly trained midwives, who were solid pillars to stand on (…) they had also shared ways to formulate basic questions*


It was also cited by many as an essential support system in cases of complex, potentially overwhelming situations, calling for team support or trust in systemic authority when outstanding measures of institutional involvement had to be called upon. This last notion was mostly contained in the case of patient 5, where an event of verbal violence, physical intimidation and threats from the patient’s perpetrator, directed at a junior doctor, came as a wave of shock throughout the entire team, and was described a posteriori as having been extremely well handled at a collective level. 


*G: I would always involve the social worker, they were the service designed to shed some light for us.*



*D: It is not easy, when you’re so young a doctor—I think she was well supported by the entire team, and the chief really stepping up into her position of authority.*


#### 10.2.4. Obstacles

As well as inspiring strategies and experience based resources, several limitations were formulated by participants on different levels. Some shared their experience of facing mental barriers of a moral nature, mostly stemming from their frustration when they felt the baby’s wellbeing was being either overlooked or mishandled.


*E: We’re all familiar with this but it is still frustrating, feeling like we would want to go way faster than the patient does*



*D: It is like I identify with the baby and beg for protection before it can.*


Others implied that their own difficulties most often arose from their own emotional sensitivity, citing empathy or emotional burdening as the main targets for adverse feelings coming from these situations. 


*I: Sometimes, I don’t know why, we just forget about it. Sometimes you feel up to asking the questions, other times we sort of brush over it when we probably linger on the question for longer.*



*It is a super difficult situation to deal with for a doctor. It is hard to witness the violence people live, it is always… well it is empathy. Maybe it is got to do with identifying with the patient.*



*F: These are painful situations, so I think I have a tendency to put them aside in my head as soon as I can, because of how psychologically burdening they are.*



*C: We all know the hardest patients situations are also the hardest to handle for HCPs*


Collective limitations were of three different natures. Firstly, several caregivers cited a detrimental aspect of workload division within a team due to the increased risk of losing information.


*A: I can’t believe this wasn’t common knowledge amongst everyone on the case, I am shocked.*



*I: Since I only focus on the medical side, I might have false information.*


This could result in repeatedly asking the same questions to the patient if the colleagues’ reported answers were insufficient or unclear, or if opportunities for questioning were limited. 


*F: Sometimes there are so many people working on the same case that I feel like if everyone brings up spousal violence over and over again, the patient might feel like they are not anything more than the abuse, and that’s counterproductive*


Finally, a negative side of a multi-professional setting was expressed by two participants, stating that when facing difficult or triggering situations, it could sometimes be tempting to do as little as possible, counting on the next consultant to fill in the blanks, always relying on the rest of the team to avoid facing hard truths they felt in no position to handle.


*F: In the end, nobody really dove into the situation because everyone was ill at ease and thought it was not their call to make because the next person would probably tackle it better.*



*F: They get so tricky that in the end nobody has a clue how to actually care for them.*


Some of these limitations were collective in nature, but institutional in essence: several participants evoked the difficulties arising from the magnitude of the hospital structure.


*A: This is a problem, the structure itself is so big that there are indeed connections between us, but they are hard to find, hard to hold on to.*


This contradicts another adverse tendency that was described twice in interviews, which is the scarcest access to interpreters in cases of language barriers, even in a Level 3 maternity ward where the migrant population is a notable part of the overall patient load. 


*C: I found there was a lot of judgemental projections in this family, and I spent the week trying to add some perspective to the mix. I thought, would it be too hard to just hold a well-intentioned attitude towards them.*



*C: It was like a runway of HCPs one after the other, just because the interpreter was here. Midwife, then pediatrician, then social worker, and I showed up in the middle of all of that.*


Lastly, one participant questioned the institutional setting of sharing information about vulnerable patients through the MPS staff, expressing their feeling that too many people were involved in these collegial meetings, most of them not having met the patient or having little to no knowledge of the case facts, but being able to partake in important decisions that did not always seem either relevant or practically feasible. 

The other aim of our study was to shed some light onto previously unnoticed points of convergence and divergence, unearthed from a variety of healthcare workers’ experiences when faced with different types of IPV. Our qualitative phenomenological analysis allowed us to isolate three main topics emerging across cases and professions.


**Understanding the initial request**


Taken chronologically, most patients’ first contact with maternity services happens during the prenatal period. It is during this timespan of varying length, depending on term of the pregnancy, that patients are likely to be oriented for assessment by their initial point of contact (mainly medical HCPs, doctors and midwives) to paramedical HCPs in order to provide the best practice care possible (social workers, psychologists) and take the necessary steps to allow the baby to be born in the best possible settings. Along these first consultations, it is likely that patients will be asked screening questions regarding the potential history of abuse they might carry; in all five of the selected cases, and reported by 7 out of 11 case-specific interviews, a shared trait of these potential abuse victims during the first consultation is their silence.

#### 10.2.5. Decrypting the Silence

The silence described can take many forms. Some participants depicted their experience of this silence as a fatalistic, implicit part of speech.


*C: It was mostly… the silences that came after anything she would share. She had a sort of… silent pout, that I interpreted a being a sign of suffering but at the same time, acceptance of a fatality that she would have to learn to live with that suffering.*



*E4: Consultations were very long because there were moments where she would share something and right after that be all withdrawn and silent, she’d look at me saying “I shouldn’t have told you”.*



*F: These patients can be in absolute mutism.*


In some cases, this silence was even perceived by participants as being corporeally expressed, interpreted from the patient’s overall demeanour, movements and occupation of space within the consultation space or the hospitalization rooms.


*F: She hadn’t set foot outside in a week, even when we offered she went for a breath of fresh air. She said, word for word “I’ll try and make it to the hallway”, when the time came for her to leave the ward. It was a way for her to express how difficult it was for her to move forward, to go outside. But it was really physical.*



*C: It was surprising, given the surgery she’d just gone through, she was on her feet really fast, she didn’t look ill at all, she just wanted out.*


In the majority of these cases, this silence was associated with a perceived absence of demand on the part of the patient, in settings where participants would have expected them to actively seek some sort of help in any regard, be it relating to the actual ongoing abuse or any other sort of assistance, from social to medical, sometimes reaching dire extremes.


*H: As time went on, she wouldn’t realize her refusal was putting the baby in danger, she couldn’t be reasoned with.*



*I: After all, delaying care for so long had gravely endangered her baby’s health, as well as her own.*


This perception of patient mutism or restraint is where another remark shared by a majority of participants over four of the selected patients originates, i.e., how well the patient really understands the role of each professional. It is a common experience for psychologists, for example, to have patients referred to them in relation to a specific risk factor or vulnerability trait, which might not have been transparently discussed with the patient, causing them to attend their psychologist appointment with little to no idea as to why the appointment makes sense in their case. In these situations, it is easily understandable that a patient unsure of the benefit a professional might provide them would remain discreet and scarcely informative, not knowing what they may obtain from sharing intimate information.


*A: From the very first consultation she alleged not seeing any interest in meeting with me.*



*C: Sometimes I try and flip the question around, I say “do you know what you’re seeking by coming here”.*


#### 10.2.6. The Balance of Benefit and Trust

Following the same logic, the interviews brought forth that providing a patient with clear, timely benefits to their current situation or quality of life drastically increases bonding between patient and caregiver, and therefore allows for more truthfulness and depth of shared information. 


*A: And providing social security rights creates some kind of trust, it directly benefits her day to day life, and that makes for more regularity in attending consultations, and over time, that’s how you feel able to tackle some questions.*



*B: The administrative side of things might not seem all that interesting, but it allows us to build a very solid patient rapport and move forward from there.*


These benefits can be material or logistical in nature, but, in several cases, have proven to be immediate in a different manner, taking root in person-to-person spontaneous proximity regardless of the healthcare setting, as depicted in the following examples, where several participants noted the impact of individual affinity as a turning point of confidence and freedom of expression. 


*F: You can feel it, when patients sort of latch onto you, you become somewhat of a reassuring attachment figure, it was obvious we had a special kind of bond*



*A: The patient was from the Caribbean, her mother was white so it is weird, but she and I sort of looked alike.*



*I: Sometimes patients feel comfortable when we ask about violence because it feels nice to them that we cared enough to ask…*


#### 10.2.7. Inauthenticity in the Patient’s Speech: Spreading Confusion or Expressing Ambivalence 

The notion of truthfulness was also mentioned several times across all patient cases, citing the likelihood of discrepancies in patient discourse according to which professional they are talking to, in relation with the feelings of gain or proximity cited above, but sometimes seemingly in the sole objective of blurring lines and raising a verbal smokescreen between the caregiving team and the abuse that has them in its grip.


*A: Discourse can be sort of empty, bland, distant, on very important subjects. (…) She could have completely different standpoints questioning the entire care she had received and I started to get worried, having this discrepancy between the real reasons why she was hospitalized and the stance she was taking in her mother’s presence.*


This discrepancy is often linked to patient ambivalence regarding the perpetrator of their abuse, understanding in part, even superficially, the demeanour of healthcare providers, more or less subtly nudging them towards self-preservation through separation and judicializing of the ordeal, and whatever social, cultural, traditional values or emotional ties are at play in keeping them in such a vulnerable position. 


*C: I think there were times where she regretted having told us some things. She actually said it clearly. “Yes, he’s very nice, forget everything I said before”*



*B: At first she didn’t want to live with him, then she changed her mind, and I wasn’t too keen on that, which she ended up holding against me in the end*


#### 10.2.8. When Keeping a Secret Reaches the Extreme

In two cases, the previously mentioned discrepancy of speech and emotional ambivalence reached their peak in increased avoidance of care to the point of terminating follow-up against medical advice, in which in one case, an addition of open aggressiveness directed at a doctor by the perpetrator. This hostility, though not as ubiquitous across cases as the other themes, has been reported by four participants in a variety of different forms, cited as follows. 


*C: The entire conversation was centred on the fact that she felt imprisoned in the ward, that she hated it, that she wanted out at soon as possible.*



*D: She had been given a mission not to get along with me.*



*D: (About the altercation between the patient’s mother and the paediatrician) I know it escalated to insults, she said “bitch”, I know the poor paediatrician was so traumatized she wasn’t sure what she had heard.*



**The baby’s existence in space and time**


The second main topic to emerge from our research was the added value of investigating the maternal investment of their future child in the context of GIPV. 

#### 10.2.9. Pregnancy and Timing

Across cases, and from personal experience reported by five participants, pregnancy appears to be a very specific moment in a woman’s life when questioning the brutality of their living conditions. 


*B: It is true that pregnancy all the way to childbirth is not a banal time in a woman’s life, it is kind of a turning point—I feel like it is all or nothing. Sometimes that’s when a woman will decide to come clean about everything, we hear that quite often, like “it is the first time I’ve ever said anything about this”.*



*E: Some patients place a lot of hope in their current pregnancy, therefore even right at the beginning of the pregnancy it’ll be time for them to confide, I think*


It also appears that specific timings during the pregnancy allow for different levels of understanding or decisiveness regarding the mother’s experience of IPV; a few participants did not recall witnessing any differences according to the term of pregnancy, although three of them evoked the second trimester of pregnancy as the timing of choice for coming to terms with the reality of IPV and the potential internal process towards making a change. 


*B: These are patients where in the beginning of the pregnancy they’ll be on their guard, they’ll still be stuck, and they are at risk of dropping out of care completely so it is complicated. And at the end they’ll be resigned, fatalistic, vulnerable with childbirth getting closer, and that’s even more complicated. So there’s sort of a period of light in the middle of pregnancy I believe.*



*D: I think the baby has to be present enough in the woman’s psyche, there needs to be transparency… I think there needs to be active foetal movement, basically.*


Although this term-specific opinion was not shared by all participants, six of them reached a point of convergence on the subject of micro-timing when it came to a patient’s decision to share their abuse situation with a professional of any sort. Micro-timing was defined by those participants as a phenomenon of a split second decision often triggered by a certain event or thought process of spontaneous, rapid occurrence. 


*B: Sometimes I think that all you need is something happening 2 s before the consultation for her to decide she doesn’t want to discuss it at all. Maybe the partner will have sent a text right then and there, so it is like all is well, do you get it? Like, it could be the right time one second, and the wrong time the next.*



*E: There might be moments of crisis which might get them to the point of filing a complaint, like that will be the right moment just like that. I don’t feel like there’s a specific timeframe more comfortable than another, apart from those crisis events.*



*D: It can be just because of that one event pushing them over the limit.*


Similarly, participants shared their experience of pregnancy as being prime change-making territory for various reasons, most having to do with a new found protective state of mind at the dawn of future maternity. This protectiveness was reported by some as being mainly directed at the baby, as an extension of maternal preoccupation.


*D: (Referring to sharing risk ratios for adverse events in children after gestational*



*IPV) With women who have a tendency to over intellectualize, it can be really helpful to share these numbers because it sort of pulls them out of their day-to-day acceptance*



*D: For her baby though, if she thinks her baby might be in danger, if she’s unhappy at home or meets that one professional with whom there’s a good rapport, I think she could try and find a way out*


Or in some cases, this protectiveness appeared to be primarily directed towards the patient themselves, either through a process of breaking the acceptance and submission cycle of abuse linked to a different self-perception during pregnancy, as is the case with patient 1, who fled her husband across national borders when her pregnancy was discovered. But in one case, this self-protectiveness seemed to arise mainly from the new birth, thus stressing the unacceptable nature of potential reproductive coercion. The newborn, in that case, was being perceived as an adverse event the mother theorized as major enough to implement a change of any matter. 


*C: Her speech was pretty transparent, that she hadn’t wanted this pregnancy, that she never wanted to be pregnant again, and that she was actually relieved that the hysterectomy had happened because that meant no more pregnancies.*


On the other hand, it appeared that, in same number of participants in three of the selected cases, the pregnancy and projection of the upcoming maternal responsibility towards the newborn-to-be was not experienced by the patient as an impulse for change, but rather a dangerous opportunity for the perpetrator to further their dynamic of overpowering and submissing the mother, under pretences of codependency and idealized family values. 


*C: In the patient’s discourse, it is often linked to a notion of not wanting to deprive their child of a fatherly presence. It has to do with these idealized representations of a mum, dad, baby nuclear family. And maybe it is also rooted in the fact that at this time, imagining themselves raising a baby alone triggers too much anxiety for them.*



*C: He had completely isolated her, she lived secluded from her family and just kept having his babies.*


#### 10.2.10. The Baby Fadeout Phenomenon

The most prominent element when exploring the baby’s psychic representation throughout cases and interviews was independent from the change-inducing potential of pregnancy and childbirth, i.e., the recurrence of “baby fade-out” in the patient discourse. This was mentioned in all cases by 9 out of 11 participants. Baby fade-out would be defined, in those cases, by the professional’s noticing the absence of the baby in the mother’s discourse or demeanor. This can come to the attention of HCPs on a very literal standpoint, through the absence or scarcity of mentioning the baby in the mother’s discourse, be it pre- or postnatal. In one postnatal case, this was expressed through the mother never mentioning the newborn by name, giving the impression of disinterest or even rebuttal.


*G5: It was something growing inside her belly. It wasn’t a being.*



*C3: I was startled by the interactions I witnessed in the mother-infant unit. She barely had any discourse about her baby, didn’t show any sign of concern about her, in a sort of denial of the needs of a premature baby who needed intention, a specific kind of care… she mostly called her “that one” instead of her name.*


In other cases, baby fadeout was also described as the tendency not to include the baby’s wellbeing in potentially life-altering decisions on the part of the mother, even in cases where HCPs explicitly recalled the importance of taking the baby into account regarding the mother’s attention on different occasions and for different reasons, ranging from maternal or pediatric health concerns to more social questionings like the new dad’s living situation. 


*B: Her daughter’s wellbeing... No, it came and went, wasn’t very present in her mind. Actually, we told her many times with the midwife, «come on, think of “baby’s name”*



*D: Parents who don’t feel guilty when something alerts the care system is always worrisome in a way, at least in the sense of negligence which still is a form of violence albeit a relatively passive form*


In most cases, neonatal, pediatric assessments revealed evidence of the selected babies being premature, underweight, requiring increased stimulation to fully thrive outside the womb; even in the first instants of a child’s life, some signs of decreased vitality and withdrawal also testify to the baby’s contextual sensory response, making it less noticeable. This was very evident in the case of patient 3, where the mother had clearly stated her disinterest in the newborn.


*C: She was a very tiny baby, who barely put on any weight, she was very sleepy. I don’t know if we can call it withdrawal at this point, but she didn’t feel very present in the moment, she wasn’t making any progress in her ability to feed, etc.*


#### 10.2.11. Noticing Maternal Override

Following that same logic, six participants over the full range of those selected mentioned the notable frequency of this baby fadeout being linked to maternal override or overpowerment. We chose the term of maternal override as a sufficiently accurate umbrella term for all phenomena where the patient’s motherhood authority is challenged by a third party, oftentimes the perpetrator of the abuse. In those cases, mothers are found to be undermined in their capability to make decisions, or even their legitimacy to do so.


*I: The grandmother doing skin to skin contact with the newborn, that’s not usual*



*C: She kept referring to a higher power that had complete authority over her (…) it was like something didn’t belong to her anymore, that could be due to her husband, there was an obvious dynamic of domination. Like being dispossessed of her own life.*



*D: She gave birth to her mother’s baby.*



**Empowerment**


The final metatheme to emerge from the interview analysis is not of the same investigative nature as the two previous one, but rather appears as an experience-based essential component of a caregiver course of action when taking on patients struggling with IPV, namely patient empowerment. Patient empowerment is the constant prerogative of both showing respect and thorough consideration for everything a patient shares or does not, in order to restore a balance where the patient is the first decision-maker within their care system. This sentiment has been described by many participants as a way to avoid unwanted reproduction of the violence dynamic already at play in their life, where one, intentionally or not, allows themselves to influence or impose their own cognitions on the patient who is already prone to submitting to another’s authority. 

#### 10.2.12. Cultural Awareness

The first step in this dynamic of giving back control to the patient is to provide the most secure environment for them to share their opinions in a morally neutral manner, backed up with a comprehensive attitude of cultural integration, which includes the caregiver’s self-awareness that they may not always be able to understand or reflect within a patient’s specific educational or traditional value-system, and in these cases, having the ability to self-analyse and withhold judgment.


*A: Really, this is something frequent with African mothers, having the family’s support, welcoming the newborn as a family with the aunt, the grandmother present*



*E: She said, “because now we’re a family” meaning that the traditional wedding implied a common living situation*


As a prolongation of this cultural awareness, supporting the endeavors of culturally inclusive caregiving within a maternity ward, allowing patients to openly share their feelings towards the care they are provided and how it fits into their personal habits, values or belief systems, seems to be a very precious tool.


*G: Sometimes during group sessions we heard mothers finally allowing themselves to say, “midwives are telling me I should bathe my baby a certain way and I don’t dare to tell them that I’ve already had four kids in my country and I always did it differently, but I’m afraid they’ll decide that I’m not capable, that I don’t do things properly just because I do things our way”.*



*E: It was obvious it meant a lot to her, to her there were principles to uphold, an order to abide by.*


#### 10.2.13. Respecting Boundaries

Working from the outside in, once a climate of mutual respect, related to cultural differences, has been established, respecting the patient’s right to intimacy is an essential tenant of respectful care. This may seem like a given in most contexts, but when dealing with cases of IPV where potentially unmistakable warning signs are spotted, being able to keep oneself in check and not cross the line of what a patient feels comfortable saying is essential for maintaining patient rapport, no matter how well-intentioned one’s insistence might be thought as justified. 


*D: I’ll always tread lightly. These are situations where if you kind of bust in, doors close and that’s very counter productive.*



*F: They are perfectly allowed not to share anything about themselves with us*


#### 10.2.14. Personal Experience over Factual Evidence

In addition to respecting a patient’s decision to share or not share certain information, respecting the patient’s viewpoint on the elements they share, or the way they choose to handle them psychologically is another crucial part of respect in an empowerment mindframe. 


*F: If it is a patient who’s doing the work, who has no issue talking about it, etc. I think we have to de-taboo the situation and collect it as we would with any other past history event*


Similarly, there is some added therapeutic value to be found in the ability to prioritize a patient’s emotional experience of their current or past situation, rather than the factual evidence HCPs might be able to gather one way or another.


*A: My intern really wanted to see what had been cited in the complaint. I thought to myself, right now with this patient, I really didn’t need to see that. I needed the patient herself to tell me what she’d gone through, how she’s experiencing things at this moment.*



*C: Sometimes I realize that if I get stuck in factual facts, that can bring up a lot of resistance on the part of the patient, or in any case a sort of “cancellation” of the events. I’d say it is better to go forward angling your questions on what the patient’s experience of the relationship, offering an ear to their feelings rather than asking them to unravel facts.*


#### 10.2.15. Giving Back Control

The final endpoint of an empowering process in the question of caring for abuse victims is to actively valorize their legitimacy in decision making on different levels. The first would be to express deference and respect to past or current choices they might already have made, in order to release them of a potential sentiment of weakness, guilt or impotency that might resemble what they might be made to feel in an abusive situation.


*E: These are people who’ve always been brought down, whose choices were never taken into account, or their desires either, and it is not our place to put them through that again.*



*F: (About deciding to give the baby up for adoption) I try and remind them that the decision they took is a mother’s decision, that it is a brave decision and that they should feel proud of themselves in their mothering role: like, being a mother doesn’t have to be the imagined scenario of giving birth and going home with a baby, sometimes it is making tough calls in order to really protect the baby. That’s acting like a proper mother.*


The other, and in most participants’ opinions, i.e., the most frequent and valuable attitude, is the respect of a patient’s use of time. That is, formulating and highlighting the fact that none of the opinions or advice given at the current time is meant to force the patient into an immediate course of action. Rather, expressing that any thought process that might have been triggered by the current care environment has all rights to carry on over all the time it needs to reach a conclusion, and reassuring the patient that whenever this conclusion is attained, the healthcare system will always be reachable and available to support and reinforce the patient in any way possible. Working at the same pace as the patient thus seems to be the best way to maintain present-time rapport during the course of the pregnancy, but also seems helpful in cementing the possibility of future change with no expiration date, a promise that the health care system is due to uphold. 


*E: We gave her all the time and space to do whatever she wanted, we said that in any case we’d always support her no matter what choice she makes.*



*F: We gave her freedom to take all the time she needed, no matter how frustrating it was for us*


## 11. Discussion

### 11.1. HCPs’ Distorted Perception of Their Quality of Care

What emerged as instantaneously noticeable during the conduction of all interviews is a tendency in healthcare providers to undermine the quality of care they provide in relation to their own challenging experience of those situations, which is consistent with previous qualitative studies of different designs [20,22]. These results are most striking in the Likert scale answers, where the clinical assessments and appreciation of severity appeared to fluctuate between HCPs involved in the same case. Through all cases, their retrospective feeling of concern for the overall situation and their self-reporting of personal efficiency and helpfulness tend to converge towards a lower average, unlike their opinion on the entire team-based care process in which they describe as more satisfactory than their own. Potential explanations for this self-deprecating phenomenon are multiple. Firstly, most HCPs reported little to no anterior training relating to IPV before joining clinical practice. This field-based expertise sometimes resonates in a self deprecating manner when thinking back on potential past situations where the caregiver feels like they could have done better, in light of what they learnt in the interim [20,22]. Secondly, there seems to be a consistent element of personal investment in these situations, allowing for crossing professional to personal boundaries like using text messaging as a means of communication with the patient instead of official exchange channels, or even in the way the HCPs justified some aspects of their caregiving decisions through personal elements like comparison with their own children or other personal life events that came to resonate with the situation at hand. This is often expressed by HCPs as a sign of lacking professionalism, which is systematically detrimental to the patient, instead of accepting the reality of them having exercised their better judgment and making a professional call in implementing more intimate elements that are relevant and pertinent within their course of action. 

Another crucial point of the misplaced impression of failure is the lack of a positive outcome criteria other than the official filing of a complaint. As most participants described, the sign of a pertinent, best-practice care process during pregnancy is often having been able to allow the patient to open up for the first time, or planting in their mind the seed of potential liberation from their abusive situation at any time in the future. This implies another purpose of pregnancy care as an entry point into self-empowerment, bearing in mind that any movement towards breaking free of the abuse might come way later, but will still be a sign of a job done right. Unfortunately, due to the magnitude of a Level 3 maternity structure, as well as the high turnover of patients, being notified of long term outcomes in these patients’ journeys is highly improbable for these caregivers, who therefore stay deprived of closure in those situations and remain wrongly convinced that they were not really of any help. This feeling might also be stressed by potential prejudice regarding available resources; many HCPs bearing a very pejorative image of child protective services and viewing their intervention as a failure, instead of integrating their involvement as a positive event of containment and socio-educational support for families that does not always result in a child’s removal from their mother’s care and subsequent trauma on all parts. 

This self-deprecating tendency in participants [22] conflicts with the obvious signs of the quality of care and efficiency shared in each of their answers. Firstly, they consistently report the feeling of patient truthfulness when enquiring about IPV, which entails a pertinent, satisfactory manner of questioning. Secondly, the perception of ambivalence or patient fluctuation showing that something inside has been shaken by the caregiver’s course of action, instead of cementing a fatalistic attitude towards the abuse cycle, attests to the depth of their dialogue and its efficiency; furthermore, the ease with which protection measures, such as secrecy and confidentiality, are put in place at the immediate service of the patients. These measures tie in with institutional organizations, namely the frequency of vulnerability staff meetings, making sure these questions are never left aside; the constant endeavor to improve transcultural approaches are evidence that whatever short term measures are accessible at any point are commonly put in place by each professional without experiencing difficulties. All of these are direct testimonies of precious, experience-based, clear-minded courses of action on behalf of all HCPs interviewed, which they do not necessarily acknowledge within themselves. A global hypothesis for this misplaced dissatisfaction would be the stark contrast between the close to absent mention of IPV during lengthy theoretical studies in France and the unbearably high frequency of those cases in clinical practice [10,20,21,22]. This is enhanced by the fact that increased experience causes for more accurate and frequent noticing of those situations [10], which may translate into a feeling of helplessness [20,22] when facing such a large scale social issue that seems ever growing, no matter how pertinent the care given to those patients is on an individual level.

### 11.2. A Dual Purpose: Creating Personal and Institutional Trust

Another important notion to arise from the entirety of the study is that any professional’s encounter(s) with a patient has a second, unsaid purpose of shaping a double alliance, interpersonal trust and rapport on one side, but overall trust in an institution on the other side; in other words, in addition to creating a climate of trust allowing the patient to confide and release themselves of their abusive burden within the consultation space, the healthcare providers are tasked with a second mission, which is convincing the patient that even beyond the timeframe of this pregnancy and the immediate rapport with her current caregivers, the institution itself stands as an unshakeable place of support and shelter at any future point in her life, being constantly available to provide her with any help she might need because supporting women in her situation is not only an individual positioning on the part of her caregivers, but also a globalized purpose on an institutional level on the part of the entire healthcare system. In order to convey that message, the institution itself has a double function to exercise. One of those is aimed inwards at the HCPs working within its boundaries, constantly providing them with up-to-date educational material, external interventions, peer-based support, an environment prone to genuine team building, and providing occasions for debate exchange. These measures should result in a collective improvement, which can be felt by every one of the HCPs and trusted to be qualitative enough to be implemented into their day-to-day practice. This is also reflected in part on the Likert scale results when comparing individual satisfaction of HCPs versus collective management satisfaction, which underlines the importance of multi-disciplinary teamwork. In a hospital of this magnitude, as some HCPs evoked on several occasions, implementing clear lines of communication, fluidity of exchanges, quality information sharing and sensible division of labor is a crucial requisite to efficient teamwork, which further stresses the internal responsibility of institutional management. The other mission is aimed towards the patients themselves and implies a close relationship to social movements and societal changes in opinions and mindsets, as well as advertising this awareness openly within the structures so as to put patients at ease, making them trust that their abuse will be handled by the entire structure to the best of their abilities, conditioned by the latest social evolutions and improvements. In some cases, cultural and traditional strings of a patient’s choice can be hard to navigate by HCPs of different upbringings, hence, it is important to keep a transcultural approach in mind in order to provide the most comprehensive and respectful care possible.

These strategies fit into a broader logic of using every possible resource to create an empowering environment for these future mothers. In heightened vulnerability cases where a form of motherhood surrender is likely to happen, the decisional power over the newborn shifts from the mother to her abuser in a mindset of self-deprecation, belittlement and resignation. Our study was consistent with the 2017 results of Chisholm et al., who suggested that this process of altered self-perception is potentially within the reach of appropriate and mindful empowerment practices on the part of HCPs, which should be systematically implemented. This empowerment course of action should be all the more valorized as it proves to be one of the safer tactics in the handling of GIPV, since engaging with the patient’s self-esteem is not as likely to trigger an adverse reaction on the part of the perpetrator as investigative screening questioning or excessive display of “escape routes” could do so.

### 11.3. Strengths and Limitations

The conducted study presents a variety of strengths that support the solidity of its findings. Firstly, our study comprises several elements underlining its external validity. Being able to interview HCPs on five very different patient cases allowed for a thorough investigation into practices in a plurality of scenarios, which allowed the study to integrate different risk factors such as a wide range of ages, cultural backgrounds, familial history, living conditions and social settings. Furthermore, conducting this study in a Level 3 maternity ward allowed us to include very critical obstetrical situations which would have been overlooked in a lesser level maternity ward where these patients would have been transferred out for critical care. 

Overall, our study presents with good internal validity; the included HCPs for each situation were those who had had the most interactions with the patients, and apart from situation 1, we were able to include most of the reported interlocutors according to patient files. The professions included in the study are also representative of the usual practices within this maternity ward, a reliable reflection of which HCPs are most likely to be called upon in situations bearing signs of intimate partner abuse. 

The number of interviews conducted in a short period of time allowed for sufficient analysis time to provide saturation of results, which were also able to be triangulated for better accuracy of IPA analysis.

Nevertheless, the study design comprises different limitations.

One limitation of our study is selection bias regarding participants. A volunteer-based design logically implies that HCPs were more likely to participate in the study if they already had a specific sensitivity or interest in the subject of IPV, which was disclosed to them beforehand, meaning that HCPs with no specific concern about IPV were not interviewed, and their own opinions or courses of actions could not be implemented in the study. That selection bias does not apply to all patient cases, since in cases 4 and 5 we were able to include all HCPs who were involved with the patients. Furthermore, no male HCPs were included in the study, meaning a potential difference in approach according to caregiver gender could not be evaluated within the study; although this limitation is more theoretical in nature and does not affect the validity of the study, given the high majority of female workers in this maternity ward and others. It is important to note that most of the preexisting literature focused on HCPs’ experience with GIPV, which tends to be focused on nurses and midwives, to which our broader inclusion of HCPs proves to be a valuable addition [22].

Finally, another limitation of the study is its monocentric design, creating a potential filter of insider politics and protocols as well as implicit systemic tensions, which could not be brought to light in comparison to another ward; nevertheless, in a qualitative design such as ours, implementing data from different institutions would have challenged our ability to properly saturate the interview content and potentially be cause for error in interpretation due to the presence of different work environments. Furthermore, with our objective of exploring implicit themes across different testimonies through IPA analysis, including different structures would have added confounding factors through referring to potentially different workplace protocols and environments, which would have added a confounding factor preventing us from accurate data saturation.

An interesting perspective for further research could be a comparative design, opposing the experience of these pregnancies by HCPs and the patients themselves [16,21], which would be the most accurate indicator of HCPs pertinence in comprehension and action. Understandably, such a design would be hard to put in place given the time-sensitive priorities of the patients that have to be respected, and would probably imply a sizeable selection bias; this speaks to the pertinence of our study design, being able to oppose different HCPs opinions on the same patient case allowing for a likely realistic extrapolation of the real-life situation.

Those findings further the preexisting need for ulterior qualitative studies, including HCPs involved in postnatal short and long term care, and call for further investigation into the articulation of maternity and pediatric care in order to sustain the potential impulses of change that might have emerged from comprehensive and thorough discussion of IPV during the entire process of bringing a new life into the world.

## 12. Conclusions

Our study was able to bring forward new and important elements of experience-based possible improvements in the pre and perinatal care of IPV victims. Theme analysis allows the sharing of strategies revolving around the patient’s comfort, confidence in the professional’s undivided, unjudgmental attention and empowerment dynamics rooted in respect, as well as well measured advice in complete transparency. In addition, the study underlines the crucial importance of trusting the institution as a reliable reference point transcending the test of time, extending beyond the pregnancy timeframe. This trust in the institution is all the more crucial in a system where healthcare workers have been and are still cited as unsatisfied by the quality of their care, relating to their individual involvement, empathy and sense of urgency as well as their own safety, which calls for an institutional support relieving its members from the depreciating concerns that appear to be unfounded on a case-by-case level.

## Figures and Tables

**Figure 1 healthcare-11-02782-f001:**
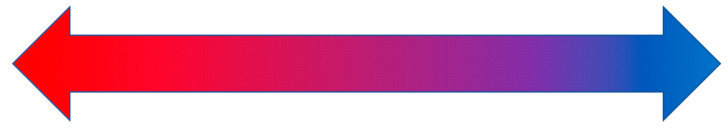
Colored visual scale marker.

**Figure 2 healthcare-11-02782-f002:**
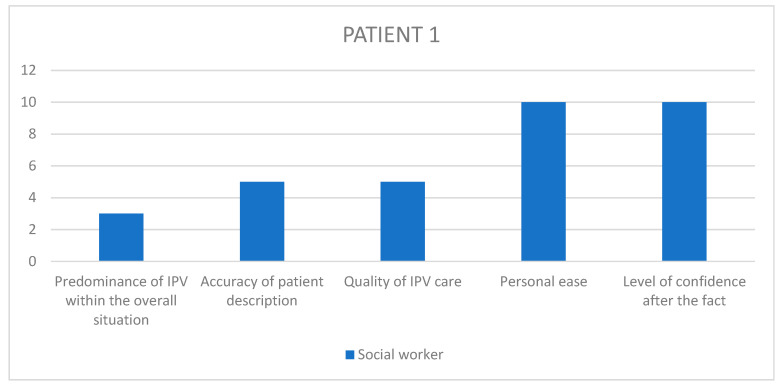
Likert scale answers for patient 1.

**Figure 3 healthcare-11-02782-f003:**
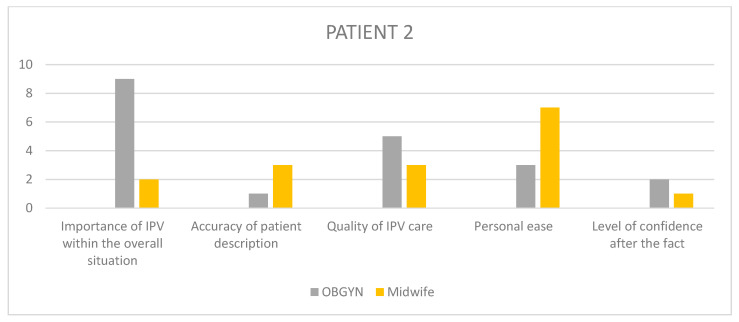
Likert scale answers for patient 2.

**Figure 4 healthcare-11-02782-f004:**
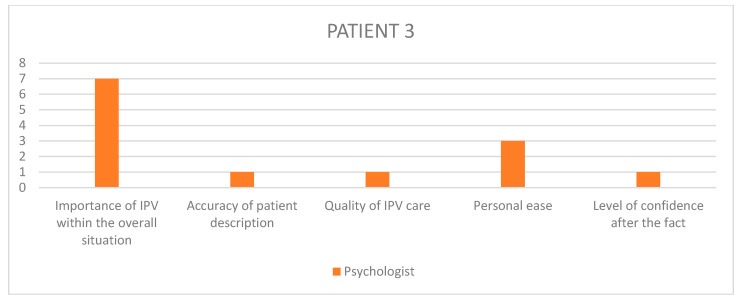
Likert scale answers for patient 3.

**Figure 5 healthcare-11-02782-f005:**
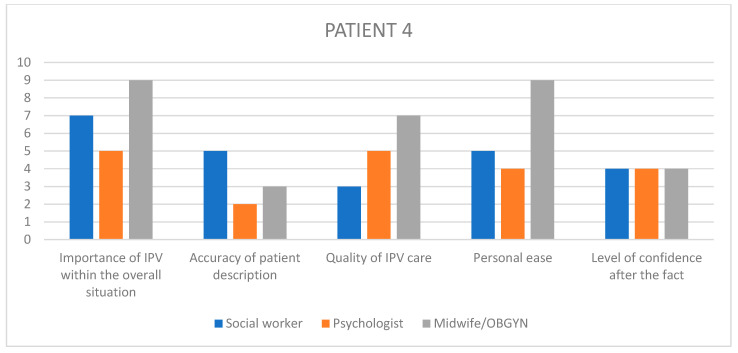
Likert scale answers for patient 4.

**Figure 6 healthcare-11-02782-f006:**
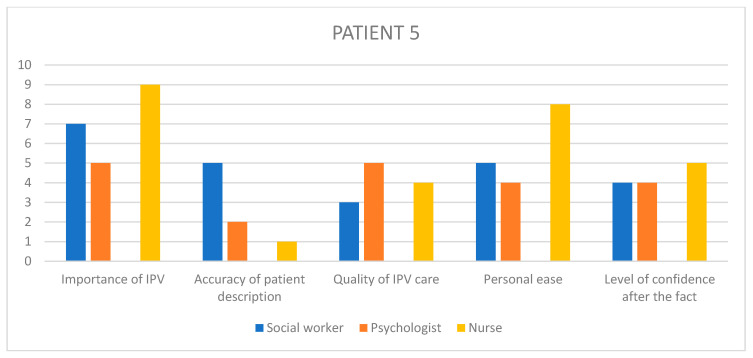
Likert scale answers for patient 5.

**Table 1 healthcare-11-02782-t001:** CDC definition of IPV subtypes.

**Physical Violence**	Includes the intentional use of physical force with the potential to cause death, disability, injury or harm.
**Sexual Violence**	Includes forcing or attempting to force a partner to take part in a sex act, sexual touching, or a non-physical sexual event (e.g. sexting) without the victim’s freely given consent, including cases in which the victim is unable to consent as a result of being too intoxicated through voluntary or involuntary use of alcohol or drugs
**Stalking**	A pattern of repeated, unwanted attention and contact with a partner that causes fear or concern for one’s own safety or the safety of someone close to the victim
**Psychological Aggression**	The use of verbal and non-verbal communication with the intent to harm a partner mentally or emotionally and/or to exert control over a partner

**Table 2 healthcare-11-02782-t002:** Interview questions.

Can you describe your job and function within the maternity ward.
How would you describe your knowledge of GIPV and its consequences? How often are you faced with such situations?
Is IPV a topic you usually bring up in consultations?
What are your personal strategies to initiate dialogue regarding potential IPV for the first time? And how do you follow up?
In your opinion, at what time of the pregnancy is it more pertinent to ask about potential IPV, in order to gather accurate information as well as securing the therapeutic relationship?
**Case-specific questions**
Can you retell the story of this patient based on your own memories? What were the most important elements?
Can you describe the context in which you first met with the patient?
What were the first signs that alerted you to the potential presence of IPV?
How could you describe the patient’s reaction to this subject?
Did you feel like you had to avoid certain questions, are there questions you wish you had asked?
How did you engage with the perpetrator, if you encountered them?
What difficulties did you encounter in this case, if any? What would you have needed to overcome them?

**Table 3 healthcare-11-02782-t003:** Likert Scale Questions (numbered 16 to 20).

GIPV was at the forefront… secondary… minimal aspect of the overall pathology
The patient initially described their experience in a minimized… accurate… exaggerated manner
In my opinion, we provided insufficient… sufficient… disproportionate attention to the violence aspect of their pathology
During the course of treatment, I felt helpless… limited… efficient in my caregiving ability
After the fact, I feel pessimistic… preoccupied… reassured regarding the future of the mother and infant

**Table 4 healthcare-11-02782-t004:** Patient characteristics at baseline.

Patient	1	2	3	4	5
Age	27	43	30	26	17
Origin	Africa	Western Europe	Eastern Europe	Africa	Mixed Caribbean
Status	Clandestine	National	Documented	Asylum seeker	National
Employment	None	NR	None	Previous	NA
Education	NR	NR	NR	University	High School
Marital status	Married separated	Separated	Married	Married	Celibate
Medical History	None	Hypertension Depression	Obesity	Excision Traumatic hearing loss	None
Reproductive history	None	Fetal death Stillbirth Prematurity Fetal death Prematurity Prematurity	7 on-term live births	None	None

NR: Not Reported. NA: Not Applicable.

**Table 5 healthcare-11-02782-t005:** Current pregnancy.

Patient	1	2	3	4	5
Desire for pregnancy	NR	Unplanned Undesired	Planned Desired	Unplanned	Unplanned
Term of discovery	14w	31w	NR	20w	12
Hospitalization	No	Yes	Yes	No	Yes
Maternal complication	No	Preeclampsia Kidney failure	P. accreta	None	Pre eclampsia
Fetal complication	None	IUGR	None	None	IUGR
Term of labor	37	31	35	41	36
Labor details	Emergency C section Hemorrhage	Emergency C section	Planned C section Severe Hemorrhage Kidney failure	Spontaneous physiological labor	Induced labor
Newborn	Healthy twins	Low birth weight Prematurity	Induced prematurity	Low birth weight	Induced prematurity Low birth weight
Postpartum	Normal	Recuperation	Recuperation	Normal	Early discharge

**Table 6 healthcare-11-02782-t006:** Violence History.

Patient	1	2	3	4	5
Author of violence	Husband (separated)	Ex-partner	Husband (current)	Husband (current) Family	Mother
Duration	18 months	Over 6 years	Unreported	Lifelong	Unknown
Type					
Physical	x	x		x	
Psychological	x	x	x	x	x
Verbal	x	x		x	
Sexual				x	
Complications		Severe obstetrical events	Isolation Withholding care	Excision Post traumatic hearing loss	Isolation Coercion Withholding care
Outcome	Fled home	Separated	Loss of touch with health care system	Separation	Loss of touch with health care system CPS notification

**Table 7 healthcare-11-02782-t007:** Participant characteristics.

	A	B	C	D	
Profession	Social worker	Social worker	Psychologist	Psychologist	
Time in the maternity	10 years	8 years	8 years	4 months	
Previous employment	No	Other ward (Non OB)	No	Yes (1 year) Other OB ward	
Initial Ipv training	Basic	Basic	No	No	
Further theoretical IPV training	Yes (thesis)	No	No	Yes (Seminars)	
Timeframe Prenatal Perinatal postnatal					
	+++	+++	+++	+++	
			++	++	
	+	++	+	++	
Met with	**Pat. 1** Prenatal	**Pat.4** Prenatal Postnatal	**Pat. 4** Prenatal Perinatal	**Pat. 5** Postnatal	
	**Pat. 5** Prenatal Postnatal		**Pat. 3** Postnatal		
	E	F	G	H	I
Profession	Midwife	Midwife	Nurse	Pediatrician	OBGYN
Time in the maternity	3 years	4 years	1 year	19 years	2.5 years
Previous employment	No	No	Yes	Unknown	Yes (part-time) Other OB ward
Initial Ipv training	Basic	Basic	No	No	No
Further theoretical IPV training	Yes (Diploma)	No	No	No	Specific training
Timeframe Prenatal Perinatal postnatal					
	+++	+++	+++	+	+
	+++	+++	+	+	+++
	++	+	++	+++	+
Met with	**Pat. 4** Prenatal Perinatal Postnatal	**Pat. 2** Perinatal Postnatal	**Pat. 5** Prenatal	**Pat. 5** Postnatal	**Pat. 2** Perinatal Postnatal

**Table 8 healthcare-11-02782-t008:** Likert Scale numerical answers—value/10—Questions 16 to 20.

	PATIENT 1	PATIENT 2	PATIENT 3	PATIENT 4	PATIENT 5
A	3	5	5	10	10																9	1	5	1	1
B																7	5	3	5	4					
C											7	1	1	3	1	5	2	5	4	4					
D																					9	1	5	1	1
E																9	7	3	9	4					
F						2	3	3	7	1															
G																									
I						9	1	5	3	2											9	1	4	6

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
