# Peer review of "Resources and Obstacles of a Maternity Staff Facing Intimate Partner Violence during Pregnancy—A Qualitative Study"

_healthcare, 2023, doi:10.3390/healthcare11202782_

Round 1
Reviewer 1 Report
please have a look in the attached file

Author Response
I have made all of the changes you pointed out and had the paper professionally grammar and spell checked. Thank you for the methodological and text composition insight.

Reviewer 2 Report
The paper reports on a qualitative study about the attitudes and behaviours of healthcare professionals in cases of interpersonal violence/domestic abuse among patients accessing maternity services. This is an important area and the findings would be an important contribution to the field.
I see no problems with the methodology and how the research has been carried out, but some with how it is reported. It seems to be presented well. I noticed a few points at which more detail could be provided, and there were substantial proofreading and grammar errors, a selection of which are noted below.
Intro:
Lines 85-86, page 3, "PTSD multiplying the risk of GIPV by a factor of 1.4" – I’m not sure whether you cover the possibility of an interactive effect, e.g. GIPV could lead to PTSD or is that screened for, i.e. only PTSD from previous trauma? also, line 90 what does "trauma-indulging mindframe" mean? That the same event will be experienced as trauma by some but not by others? More should be done to emphasise that you are not saying that victims are culpable in their own abuse.
Setting: Level 3 maternity ward - this is mentioned five times in the paper but at no point is it explained what this means for an international audience. From context, I can see that it's a critical care environment, but could there be a short description of what the levels are and what the differences between a 1, 2, and 3, and if there is a level 4?
Methods:
Lines 287 and 294, p.9 - 9 healthcare providers and 8 generic interviews, is this a typo or did one of the HCPs not do a generic interview? This should be highlighted if so.
Analysis:
PP 11-12, in the bar charts, the legends indicate the HCPs who answered for each patient, but these aren't always present in the results - e.g. patient 2 has social worker, psychologist, OBGYN, midwife, but only the OBGYN and midwife results are present. If those participants didn't give a score, would it make more sense that they are excluded from the legend.
There should be an overall proofread and grammar check because there are some places where the grammar doesn’t seem quite right, or there seems to be a missing word or the wrong word is used. I’ve tried to detail all of those I spotted below but there are more than these listed.
I suggest the proofread/grammar check also includes the quotes in the results section as some of the quotes seem to have unusual grammar - have these been translated (assuming they were conducted in French)?
Intro - line 44, p.1 "(including coercive tactics)" is repeated within a sentence.
Line 54, p.2, “Table 1: CDC definition of IPV subtypes - sexual violence” appears unfinished, as it says 'any of the following acts' and then doesn't list acts.
Results - lines 397-399, p.14 is a repeated quote from the previous page, lines 373-375. Suggest leave it in at 373 and remove from 397 as there are more quotes to illustrate your point about warning signs on p.14.
Quotes sometimes have a colon between the participant ID (A, B, C etc.) and sometimes not. In some there's a forward slash (line 425), or a period (line 473, 487). Some have a larger space between the participant ID and the quote. There should be consistency in the style of presentation.
Line 511, p.16, and line 840, p.22 "wholesome care" could mean "holistic care", or does it mean "best practice care"? Either makes sense in context.
Line 515, p.16 "reported by 7 out of 11 professionals" - there were only 9 professionals. This could be rephrased to indicate that it was reported in 7 out of the 11 case-specific interviews (if that's what is meant). Also happens in line 686, p.19 "9 out of 11 participants".
Line 555, p.17 quote is attributed to A5 - is the 5 a typo? No other quotes are attributed to a letter-number combination up to this point, but then increasingly there are more quotes attributed to a letter/number combination. Also F2 at line 573, and A5 at 575 - are these in relation to the patient number? I’m guessing that it is that the quotes with just a letter are the generic interviews with HCPs and those with a number are the client-specific? this should be made more clear as when it starts happening it comes across as a typo.
Quote marks are sometimes the English versions “” and sometimes the French « » - with line 595 using both in the same quote “Yes, he’s very nice, forget everything i said before »
Line 846, p.22 "journey's" should be "journeys"
Line 849, p.23 "hcps" should be "HCPs"
Line 846, p.23 "proof" better phrasing would be "evidence"
Line 889, p.23 "bound" does this mean "boundaries"?
Author Response
I have completed the entirety of the corrections you pointed out in my final manuscript and had the final version grammar and spellchecked professionally. Many thanks for your input. please see attachment.

Round 2
Reviewer 1 Report
please have a look at the file
